# LINEAR-TIME SEQUENCE MODELING WITH MLPS

## ABSTRACT

We present Causal Relation Networks (CausalRNs), the first all-MLP sequence modeling architecture with linear-time parallel training. To enable autoregressive modeling, we made Relation Networks (RNs) equivariant and causal through relaxation and masking. Contrary to the earlier belief that RNs are quadratic-time, we show that when using $\exp(x)$ as the activation function, any RN is linear-time, fully parallelizable, and numerically stable. Our derivation spontaneously gave rise to familiar design choices adopted by state-of-the-art architectures, e.g. exponential gating and state expansion. Such duality provided a new perspective, from which we not only validated popular design choices, but also discovered new design considerations. Experiments on autoregressive language modeling and image classification showed CausalRNs to be comparable to Linear Transformers. The quadratic variant of CausalRNs achieved perfect retrieval on the copying task, which was previously only possible with Transformers.

## 1 INTRODUCTION

One of the mysteries in the field of sequence modeling is how the Transformer architecture has remained largely the same since its inception in 2017 (Vaswani et al., 2017). While the field has seen tremendous progress—from successfully scaling up these architectures in language modeling (Radford et al., 2018; Achiam et al., 2023) to realizing that most machine learning tasks can be formulated as instances of sequence modeling (Carion et al., 2020; Sharir et al., 2021; Achiam et al., 2023)—our theoretical understanding of Transformers remained lacking.

Contrasting this limited understanding of Transformers is our increasing knowledge of Multi-Layer Perceptrons (MLPs) (Rosenbaltt, 1957), the fundamental building blocks of deep learning. Over the years, we have gained increased insights into MLPs, from the theory of Neural Tangent Kernels (NTK) (Jacot et al., 2018) to the empirical and theoretical guarantees of global convergence in the over-parameterized regime (Barboni et al., 2022). MLPs' lack of strong inductive biases has also led to the emergence of all-MLP architectures that offer interesting insights. For example, the MLP-Mixers (Tolstikhin et al., 2021) showed that MLPs can be powerful image classifiers, and directly scaling MLPs (Bachmann et al., 2024) demonstrated their efficiency and scalability. However, there has not yet been a Transformer-style all-MLP architecture that supports autoregressive modeling.

In this paper, we propose a novel all-MLP (Rosenbaltt, 1957) architecture called the Causal Relation Network (CausalRN) (see Figure 1). The original Relation Networks (RNs) were proposed to model the complex pairwise relationship within a set of feature vectors, but had limited scalability, flexibility, and computational efficiency (Santoro et al., 2017). In fact, training an RN requires computing a quadratic number of shared-weight MLPs, with respect to the number of feature vectors. RNs are also incompatible with autoregressive modeling.

To make RNs compatible with autoregressive sequence modeling(Vaswani et al., 2017; Radford et al., 2018), we made two key modifications. First, we relaxed one of their summation steps. This makes RNs equivariant like Transformers and enables stacking of RN blocks to create deeper networks. Second, we causally masked the summation step (see Figure 1 (d)), which is a requirement for models to train and predict autoregressively (Radford et al., 2018).

Each CausalRN block computes a quadratic number of MLPs, which can be computationally heavy. We discovered that the input and output layers can be pre-computed in linear time and reused. In addition, we show that CausalRNs can be fully linearized simply by switching the activation function from ReLU (Nair & Hinton, 2010) to the exponential function, $\exp(x)$. This linearization is exact,

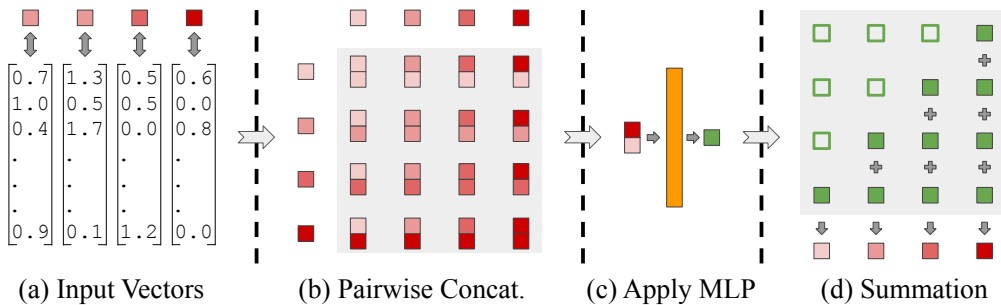

(a) Input Vectors  (b) Pairwise Concat.  (c) Apply MLP  (d) Summation

Figure 1: Illustration of a CausalRN block. Each square is a feature vector. (a) First, we arrange a number of input vectors in sequence. (b) Second, the vectors are pairwise concatenated with each other. (c) We then apply an MLP to each one of the concatenation. (d) Finally, we add the vectors along an axis to obtain the output vectors. This step is causally masked.

i.e., no approximations are needed. All exponential computations can be carried out in the log domain with the log-sum-exp trick. As a result, CausalRNs have some desirable properties. During training, CausalRNs are linear time, fully parallelizable, and numerically stable. At inference time, CausalRNs support $O(1)$ streaming, similar to State Space Models (SSMs) (Gu & Dao, 2023).

Our ablation study uncovered deep connections between CausalRNs and state-of-the art architectures. Specifically, using $\exp(x)$ as the activation function is similar to exponential gating (Beck et al., 2024; Yang et al., 2023), and the fact that MLPs have wide hidden states is closely related to the idea of state expansion found in SSMs and Linear Attention architectures (Gu et al., 2022; Zhang et al., 2024). We confirm the importance of these popular design choices.

A surprising discovery emerged when we force CausalRNs to become quadratic. By using pre-activation normalization, we recover an infinitely growing matrix-valued state that is irreducible to a single vector. We hypothesize that this allows for efficient in-context retrieval, similar to the KV cache of a Transformer (Vaswani et al., 2017). Our results suggest matrix-valued states are critical for designing future architectures that might try to match or surpass Transformers.

This work is a scientific investigation in machine learning (Nakkiran & Belkin, 2022). Our main contribution lies in revealing novel theoretical results and surprising phenomena. In the process, we raise new research questions and propose new research directions. We do not position the CausalRN as a replacement for Transformers or State Space Models (SSMs).

## 2 PRELIMINARIES

**Notation** We use bold lowercase letters for vectors and bold uppercase letters for matrices. $[\mathbf{x}_i; \mathbf{x}_j]$ represents the vertical concatenation of $\mathbf{x}_i$ and $\mathbf{x}_j$. We use $\circ$ for element-wise product. To describe the size of a neural network, we use $d_e$ for embedding size and $d_h$ for the number of hidden neurons.

**Multi-Layer Perceptrons (MLPs)** Multi-layer perceptrons (MLPs) (Rosenbaltt, 1957) are the simplest form of a neural network. MLPs were originally introduced as a method to fit non-linear functions for data with vector inputs and outputs. They emerged as powerful architectures in machine learning due to their ability to approximate any continuous function (Hornik et al., 1989). The single-hidden-layer MLP is a foundational building block for more complex neural network architectures (Zaheer et al., 2017; Santoro et al., 2017; Vaswani et al., 2017).

**Definition 2.1** (Single-hidden-layer MLP Module). Consider a non-linear element-wise activation function $\psi$. For input $\mathbf{x} \in \mathbb{R}^{d_e}$ and parameters $\mathbf{b}_{in} \in \mathbb{R}^{d_h}, \mathbf{b}_{out} \in \mathbb{R}^{d_e}, \mathbf{W}_{in} \in \mathbb{R}^{d_h \times d_e}$, and $\mathbf{W}_{out} \in \mathbb{R}^{d_e \times d_h}$, a single-hidden-layer MLP is defined as

$$f_\theta(\mathbf{x}) = \mathbf{W}_{out}\psi(\mathbf{W}_{in}\mathbf{x} + \mathbf{b}_{in}) + \mathbf{b}_{out} \tag{1}$$

Despite their remarkable properties, a critical limitation of MLPs is that they can only accommodate feature vectors with fixed dimensions (Zaheer et al., 2017). This inflexibility limits their utility for sequential modeling where models must adapt to varying feature dimensions.

**Deep Sets**   Deep Sets (Zaheer et al., 2017) were introduced as a method to model a variable number of features using MLPs (Zaheer et al., 2017). Deep Sets sum the outputs of a shared-weight MLP applied to individual feature vectors. Notably, (Zaheer et al., 2017) proved that summation of feature vectors preserves the universal approximation property.

**Definition 2.2** (Deep Sets Module). For a set of inputs $X = \{\mathbf{x}_1, \mathbf{x}_2, ..., \mathbf{x}_n\}$ where $\mathbf{x}_i \in \mathbb{R}^{d_e}$, and $f_\theta : \mathbb{R}^{d_e} \to \mathbb{R}^{d_e}$, a single *Deep Sets Module* is define as

$$\mathbf{y} = \frac{1}{n} \sum_{i=1}^{n} f_\theta(\mathbf{x}_i) \tag{2}$$

Although the Deep Sets module can be followed by another MLP, we cannot stack multiple layers of Deep Sets modules, because the output is a single vector.

**Relation Networks**   Relation Networks (RNs) (Santoro et al., 2017) can been seen as an extension to Deep Sets. The goal was to explicitly model the dependencies between features vectors, similar to Transformers (Vaswani et al., 2017). A major strength of RNs is their simplicity: they only involve MLPs and summations of vectors.

**Definition 2.3** (Relation Network Module). For a set of inputs $X = \{\mathbf{x}_1, \mathbf{x}_2, ..., \mathbf{x}_N\}$ $\mathbf{x} \in \mathbb{R}^{d_e}$, and $f_\theta : \mathbb{R}^{2d_e} \to \mathbb{R}^{d_e}$, a single *Relation Network Module* is defined as

$$\mathbf{y} = \frac{1}{n^2} \sum_{i=1}^{n} \sum_{j=1}^{n} f_\theta([\mathbf{x}_i; \mathbf{x}_j]) \tag{3}$$

While RNs serve as a promising starting point for our search, they have a few significant downsides. First, like Deep Sets, RN modules are not stackable. Second, it is not immediately obvious how RNs can perform autoregressive modeling. Finally, applying MLPs to each vector concatenations make RNs challenging to scale up. We address all these concerns in the following section.

## 3 CAUSAL RELATION NETWORKS

In this section, we describe how we reformulated and modernized Relation Networks (Santoro et al., 2017) to create the Causal Relation Network (CausalRN) architecture. In Section 3.1, we describe the foundational architectural change that makes CausalRNs equivariant and causal, similar to decoder-only Transformers (Radford et al., 2018). In Section 3.2, we show the surprising linearizability of exponentially-activated Relation Networks. In Section 3.3 and Section 3.4, we introduce our design choices when it comes to normalization layers and discuss their impact.

### 3.1 EQUIVARIANCE AND CAUSALITY

A single Relation Network (RN) block takes in a set of feature vectors and outputs a single vector (Santoro et al., 2017). This makes it impossible to apply residual connections (He et al., 2016) or stack into multiple layers.

We found that relaxing the inner summation of Eq. 3 makes an RN equivariant. This permits us to stack RN blocks and apply residual connections.

**Definition 3.1** (Bidirectional Relation Network (BiRN) Module). For a set of inputs $X = \{\mathbf{x}_1, \mathbf{x}_2, ..., \mathbf{x}_n\}$ where $\mathbf{x}_i \in \mathbb{R}^{d_e}$ and $f_\theta : \mathbb{R}^{2d_e} \to \mathbb{R}^{d_e}$, a single *Bidirectional Relation Network Module* is define as

$$\mathbf{y}_j = \frac{1}{n} \sum_{i=1}^{n} f_\theta([\mathbf{x}_i; \mathbf{x}_j]) \tag{4}$$

BiRNs are analogous to Bidirectional Transformers such as BERTs (Kenton & Toutanova, 2019), since each output vector aggregates information from both past input vectors and future input vectors. With a placeholder token CLS, BiRNs can perform the classification tasks (see 4.3).

For Relation Networks to perform sequence modeling, another architectural property that we need is causality. Causally-masked architectures allow for efficient autoregressive training (Vaswani et al., 2017). This can be achieved by capturing the causal subset of all relationships between feature vectors (see Figure 1 (d)).

**Causal Relation Network Module** For a set of inputs $X = \{\mathbf{x}_1, \mathbf{x}_2, ..., \mathbf{x}_n\}$ where $\mathbf{x}_i \in \mathbb{R}^{d_e}$, and $f_\theta : \mathbb{R}^{2d_e} \to \mathbb{R}^{d_e}$, a single *Causal Relation Network Module* is defined as

$$\mathbf{y}_j = \frac{1}{j} \sum_{i=1}^{j} f_\theta([\mathbf{x}_i; \mathbf{x}_j]) \tag{5}$$

CausalRNs are analogous to Causal Transformers such as GPTs (Radford et al., 2018), since each output vector $\mathbf{y}_j$ can only aggregate information from past vectors $\mathbf{x}_{1 \sim j}$. Their causal nature makes them suitable for auto-regressive sequence modeling.

## 3.2 EXPONENTIALLY-ACTIVATED RELATION NETWORKS ARE TIME AND SPACE EFFICIENT

Relation Networks (Santoro et al., 2017) use MLPs (Rosenbaltt, 1957) to model the pairwise relationships between feature vectors. However, implementing a quadratic number of MLP modules can be computationally prohibitive. We made a surprising discovery that by using the exponential function as the activation function, all Relation Networks introduced so far can be linearized, including the vanilla RN. This linearization is exact, meaning no approximations are needed. We start with the key insights for BiRNs.

**Proposition 3.2.** *Exponentially-activated Bidirectional Relation Networks have $O(N)$ time and space complexity for $N$ inputs.*

*Proof.* Consider a set of inputs $X = \{\mathbf{x}_1, \mathbf{x}_2, ..., \mathbf{x}_N\}$, let $f_\theta$ be a single-hidden-layer MLP from Eq. 1. Expanding Eq. 4, we get

$$\mathbf{y}_j = \frac{1}{N} \sum_{i=1}^{N} \left( \mathbf{W}_{out} \psi(\mathbf{W}_{in}[\mathbf{x}_i; \mathbf{x}_j] + \mathbf{b}_{in}) + \mathbf{b}_{out} \right). \tag{6}$$

Split $\mathbf{W}_{in}$ into $(\mathbf{W}_{left} \ \mathbf{W}_{right})$, and moving the affine mapping of $W_{out}$ to the outside, we have

$$\mathbf{y}_j = \mathbf{b}_{out} + \frac{1}{N} \mathbf{W}_{out} \sum_{i=1}^{N} \psi(\mathbf{W}_{left}\mathbf{x}_i + \mathbf{W}_{right}\mathbf{x}_j + \mathbf{b}_{in}). \tag{7}$$

Denote that $\mathbf{p}_i = \mathbf{W}_{left}\mathbf{x}_i$ and $\mathbf{q}_j = \mathbf{W}_{right}\mathbf{x}_j + \mathbf{b}_{in}$, then we can focus on the summation step:

$$\sum_{i=1}^{N} \psi(\mathbf{p}_i + \mathbf{q}_j) \tag{8}$$

We notice two things. First, we can pre-compute all $\mathbf{p}_i$ and $\mathbf{q}_j$ in $O(N)$ time. Second, the output affine mapping can be deferred until after the summation step. This means that the input and output layers of a single-hidden-layer MLP need not be involved in any quadratic operations.

Setting $\psi$ as the exponential function, by the exponential property,

$$\sum_{i=1}^{N} \exp(\mathbf{p}_i + \mathbf{q}_j) = \sum_{i=1}^{N} \exp \mathbf{p}_i \circ \exp \mathbf{q}_j = \exp \mathbf{q}_j \circ \sum_{i=1}^{N} \exp \mathbf{p}_i \tag{9}$$

This means that we can pre-compute and reuse $\sum_{i=1}^{N} \exp \mathbf{p}_i$ for any $j$. $\qquad \square$

We refer the readers to Appendix A for numerically stable implementations of both CausalRNs and BiRNs. The key insight is to move the exponential computations into the log domain and use the log-sum-exp trick. We test the performance of BiRNs and CausalRNs on Image Classification and Language Modeling respectively (see Section 4.3).

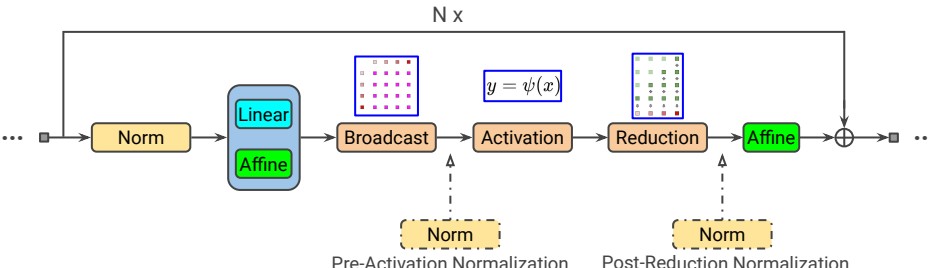

Figure 2: The computational pipeline of a single CausalRN block. A set of vectors goes through Pre-Normalization (Xiong et al., 2020), which is then converted into two sets of vectors using linear and affine mappings. During Broadcast, the vectors from the two sets are summed pairwise. In Activation, we apply the activation function to each vectors. During Reduction, all vectors are summed along a given axis. Finally, we perform an output affine mapping and a residual connection. The dashed lines describe candidate locations to insert normalization layers. Any pre-processing and post-processing steps are omitted on both ends, which typically involve embedding layers and classification layers.

### 3.3 POST-REDUCTION NORMALIZATION

In its original definition, a Relation Network (Santoro et al., 2017) uses summation to aggregate information from feature vectors (see Eq. 3). To ensure the resultant vector has a stable variance, we propose post-reduction normalization (see Figure 2). This is achieved by applying Layer Normalization (Ba et al., 2016) right after the summation step. We test post-reduction normalization and its effect in Section 4.2.

As added benefits, post-reduction normalization allows for better interpretability. We visualize an attention map extracted from a vision BiRN in Section 4.4.

### 3.4 PRE-ACTIVATION NORMALIZATION

We showed that exponentially-activated CausalRNs are reducible to linear time and space complexity. This is beneficial from an engineering point-of-view, but it could hurt in-context retrieval ability (Oren et al., 2024; Wen et al., 2024). This is because a linearized CausalRN has a fixed vector-valued memory. In contrast, a Transformer has an infinitely growing matrix-valued memory (Oren et al., 2024), commonly known as a KV cache (Vaswani et al., 2017).

Different from RNNs (Peng et al., 2023) and SSMs (Gu & Dao, 2023), CausalRNs offer an elegant way to induce an infinite matrix-valued memory. By applying pre-activation normalization (see Fig. 2), exponentially-activated CausalRNs become irreducible. This is because the exponential property no longer holds for $\exp\left(\mu(x + y)\right)$, where $\mu$ is a normalization operator. They also take quadratic time to train, so we introduce an approximation, $\exp\left(\mu(x) + \mu(y)\right)$, which preserves the exponential property and linearizability. We call this the approximate pre-activation normalization.

We hypothesize that, while a generic CausalRN maintains an infinitely growing matrix-valued memory, it is degenerate and cannot hold as much information. Pre-activation normalization induces a non-degenerate infinite matrix-valued memory that enables effective in-context retrieval, similar to a KV cache (Vaswani et al., 2017). We test this hypothesis in Section 4.4.

## 4 EMPIRICAL RESULTS

### 4.1 SETUP

In this section, we present our empirical evaluation of the proposed CausalRN architecture. We conduct an extensive ablation study to validate our design choices (Section 4.2), demonstrate the effectiveness of CausalRN on character-level language modeling using the WikiText-103 dataset (Section 4.3), and perform transfer learning from CIFAR-5M(Nakkiran et al., 2021) to CIFAR-10

and CIFAR-100 for image classification (Krizhevsky et al., 2009) (Section 4.3). We also include experiments on the copying task to assess the model's ability to handle long-range dependencies (Section 4.4), and provide interpretability results to highlight the attention-like behavior of BiRN (Section 4.4). For full implementation details, we refer readers to Appendix A.

**The Copying Task**  The copying task requires a language model to repeat a given random string, testing its memory capacity and in-context learning ability, especially when the string grows exponentially in length. We configure the copying task experiment in the same way as (Jelassi et al., 2024). We uniformly choose each character from the alphabet. There are three special tokens: `<BOS>` to signal the beginning of a random string, `<SEP>` to signal the end of the random string, and `<EOS>` to signal the end of the generation process. In our experiments, the string length ranges from 16 to 256, thus the corresponding context window ranges from 34 to 514.

## 4.2 ABLATION STUDY

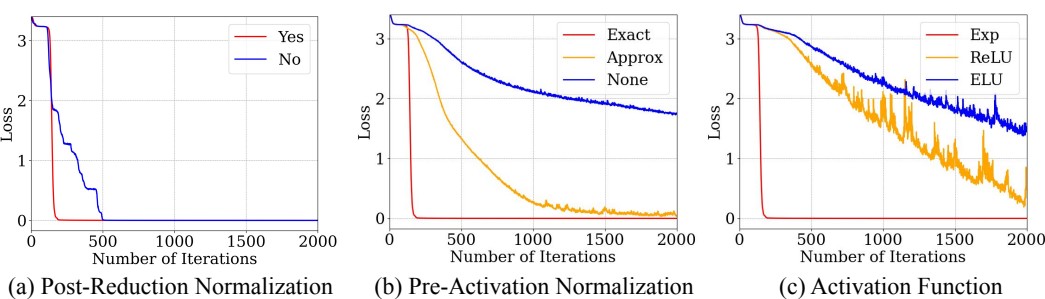

(a) Post-Reduction Normalization  (b) Pre-Activation Normalization  (c) Activation Function

Figure 3: Ablation study results. The red lines correspond to our proposed CausalRN configuration. The blue or orange lines indicate the removal or changing of one of the components.

We performed a careful ablation study to evaluate the contribution of our proposed components. All experiments are performed on the copying task with a string size of 128.

In Figure 3 (a), we see that post-reduction normalization improves the training stability. This verifies our assumption that normalization after vector summation stabilizes CausalRNs.

In Figure 3 (b), we observe that only exact pre-activation normalization induces an obvious phase change phenomenon (Nanda et al., 2023) near the 200[th] iteration. This highlights the importance for sequence modeling architectures to maintain matrix-valued memory states.

In Figure 3 (c), we see that the use of the exponential activation function fundamentally accelerates convergence, while both ReLU and ELU show gradual and linear descent. A possible explanation is $\exp(x)$ makes it easy for important tokens to dominate the feature vector after the reduction step. This phenomenon reminds us to re-examine the role of $\exp(x)$ in architecture design.

In Figure 4 (d), we increase the number of hidden neurons in each MLPs by a state expansion factor. As the factor increases, the model's memorization capacity also grows. We observe an early convergence phenomenon, suggesting sequence modeling architectures may not require infinite memory states to achieve perfect retrieval.

The ablation study provided critical insights into the effectiveness of each component within the CausalRN architecture. We validated popular design choices such as exponential gating and state expansion. We discovered the importance of proper normalization, the role of a matrix-valued memory state, and the early convergence phenomenon for state expansion. Understanding the role of each component is essential for designing future architectures.

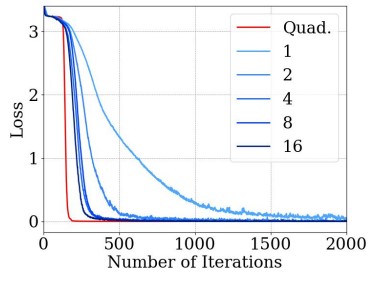

(d) State Expansion

Figure 4: State expansion results.

## 4.3 QUANTITATIVE RESULTS

We conducted a quantitative evaluation of our proposed architectures, the linear CausalRNs and BiRNs, against several baselines on three benchmarks: WikiText-103, CIFAR-10, and CIFAR-100. The performance metrics used were Perplexity (PPL) for WikiText-103 and Accuracy (Acc) for CIFAR-10 and CIFAR-100.

For language modeling, we use Linear Causal-RNs to implement a character-level language model. We use WikiText-103 dataset, which is a language modeling corpus derived from Wikipedia articles. We use a ASCII-based character-level tokenizer with a vocabulary of 128. After character-level tokenization, the training set totals 522,243,436 tokens. This allows the model to converge within a single

Table 1: Performance comparison on Wikitext-103. PPL: Perplexity (lower is better).

| Model | WikiText-103 PPL $\downarrow$ |
|---|---|
| Transformer | 2.70 |
| Linear Transformer | 2.81 |
| Mamba-1 | **2.52** |
| **Linear CausalRN (ours)** | 3.22 |

epoch. We train the model using 320 batch size for one epoch. For optimization, we use Adam with $1 \times 10^{-4}$ learning rate. As shown in Table 1, although Linear CausalRN does not surpass competitive architectures such as Transformers (Vaswani et al., 2017) and Mamba (Gu & Dao, 2023), they are still valid ways to perform autoregressive sequence modeling.

For image classification, we use Linear BiRNs to implement a vision model. We added a CLS token to output the classification result. Positional embeddings are randomly initialized and trainable. They are added to the initial feature vectors. The dataset we choose is CIFAR-5M (Nakkiran et al., 2021). CIFAR-5M is an extended version of CIFAR-10 (Krizhevsky et al., 2009), including 5 million synthetic images. The abundance in data allows the model to converge within one epoch. We train the model using 320 batch size for one epoch. For optimization, we use Adam with $5 \times 10^{-4}$ learning rate. All models are later fine-tuned on CIFAR-10 and CIFAR-100, to demonstrate the transferability. As shown in Table 2, the Linear BiRN is able to classify images with reasonable accuracy. While not competitive, they are valid ways to perform bidirectional sequence modeling.

## 4.4 QUALITATIVE RESULTS

**Comparing Learning Curves** In Figure 5, we set the string size to 128 and directly compare the learning curves of the quadratic CausalRN and Transformer. We notice that the curves are very similar, even after converging to near zero loss values. This implies that Causal-RNs might have some underlying connection with Transformers. The initial plateau and near

Table 2: Performance comparison on CIFAR-10 and CIFAR-100. Acc: Accuracy (higher is better).

| Model | CIFAR-10 Acc (%) $\uparrow$ | CIFAR-100 Acc (%) $\uparrow$ |
|---|---|---|
| Transformer | **84.87%** | **50.56%** |
| Linear Transformer | 80.90% | 48.64% |
| **Linear BiRN (ours)** | 78.57% | 42.10% |

vertical descents from both models is a clear indication of the phase change phenomenon (Nanda et al., 2023).

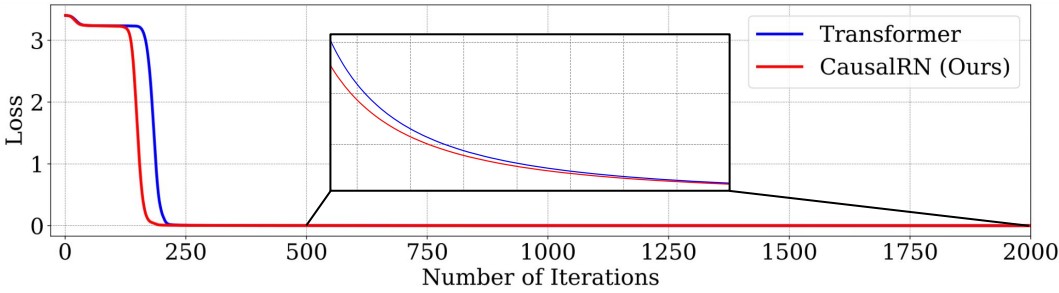

Figure 5: Comparison of learning curves. We zoom in from 500 to 2000 for closer observation.

**Comparing Convergence** In Figure 6, we vary the string sizes from $2^4$ to $2^8$ to observe the scaling property of four models. Notably, simply by adding pre-activation normalization, CausalRNs change from hardly converging to converging faster than Transformers. The linear models whose memory is reducible to fixed vectors constantly perform worse than their quadratic counterparts, aligned with prior observations (Jelassi et al., 2024). This result verifies our claim that applying pre-activation normalization recovers an irreducible infinite matrix-valued memory, which in turn support effective in-context retrieval.

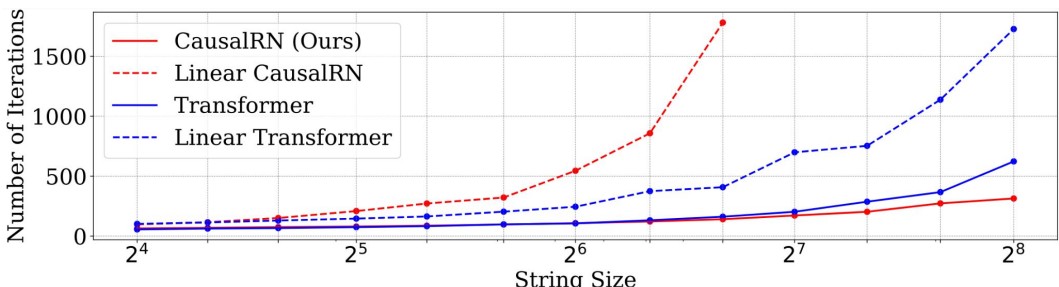

Figure 6: Convergence comparison. The y values correspond to the number of iterations until the model reaches 99% accuracy. Linear CausalRN did not converge for string sizes $\geq 2^7$.

**Interpretability Analysis** In Figure 7, we visualize heat maps extracted from a trained Linear BiRN model. The heat maps come from the 6th layer and show how strongly each patch attends to the `<CLS>` token. It demonstrates that the model is able to focus on the main object and ignore background elements. Specifically, the left image shows the original frog image, and the right image displays the corresponding attention heatmap. The heatmap indicates that the model focuses primarily on the main object. For example, for the lower image, the heatmap indicates that the model focuses primarily on the plane, effectively ignoring the sky and ground.

The visualization provides insights into the model's attention mechanism, highlighting its ability to differentiate between relevant and irrelevant features within an image.

## 5 DISCUSSION

**Conclusion** In this paper, We proposed Causal Relation Networks (CausalRNs), the first all-MLP architecture that supports autoregressive sequence modeling. The combination of similarities and fundamental differences from existing architectures, such as Transformers and SSMs, helped us validate popular design choices and discover new design considerations. Our findings highlight the importance of exponential gating, state expansion, and a matrix-valued memory state for effective in-context retrieval. Moreover, our results suggest that Transformers' specific construct may not be the only way to excel at in-context retrieval.

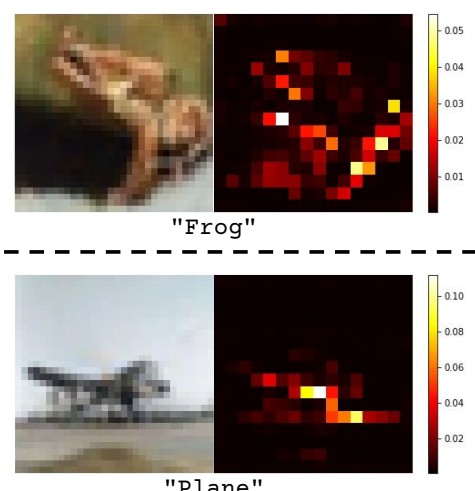

Figure 7: Interpretability result showing both the original images and the corresponding heat maps.

**Limitations** We do not expect CausalRNs to replace Transformers in machine learning application, due to come clear limitations: (1) CausalRNs do not have a multi-head scheme; (2) the current implementation, while linear-time and parallel-trainable, is not I/O-aware with wide hidden layers; (3) CausalRNs do not fully utilize tensor cores. Nonetheless, CausalRNs are valuable for theoretical explorations, and is very computationally practical.

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

# A   THE NUMERICALLY STABLE IMPLEMENTATIONS

It is straightforward to implement Bidirectional Relation Networks (BiRNs) and Causal Relation Networks (CausalRNs) in PyTorch (Paszke et al., 2019). In Figure 8, we share code snippets for the key computation of of both modules. Variable `a` and `b` each has shape (`batch_size`, `num_tokens`, `emb_size`). Variable `a` corresponds to the set of vectors outputted by a linear layer, and variable `b` corresponds to the set of vectors outputted by an affine layer. The key insight is to use `logsumexp` or `logcumsumexp` to perform the reduction step before subtracting maximum values along certain axes. This stability trick preserves collinearity. Therefore, as long as we apply post-reduction normalization, our stability trick is exact. The full code will be release on github.

```python
def linear_bidirectional_stable(a, b):
    b = b.logsumexp(dim=1, keepdim=True)
    const_b = b.max(dim=1, keepdim=True)[0].detach()
    b = b - const_b
    a = a + const_b
    const_a = a.amax(dim=2, keepdim=True).detach()
    a = a - const_a
    return a.exp() * b.exp()
```

```python
def linear_causal_stable(a, b):
    b = b.logcumsumexp(dim=1)
    const_b = b.cummax(dim=1)[0].detach()
    b = b - const_b
    a = a + const_b
    const_a = a.amax(dim=2, keepdim=True).detach()
    a = a - const_a
    return a.exp() * b.exp()
```

Figure 8: Code snippets for implementing numerically stable BiRNs and CausalRNs.

# B   IMPLEMENTATION DETAILS

**Architecture**   We compared our CausalRNs with Transformers (Vaswani et al., 2017), Linear Transformers(Katharopoulos et al., 2020), and Mamba (Gu & Dao, 2023). There are two scale variants: base and tiny. For base models, we set $d_e = 768$ and $d_h = 3072$. The number of layers is 12. The number of heads is 12, if applicable. Under the base setting, both CausalRNs and Transformers have 85.6M parameters, and the Mamba architecture has 45.8M parameters. For tiny models, to match the scale of the copying task, we set $d_e = 192$, which was 150% of 128 and 75% of 256. 128 and 256 are the length of the random sequence that we are copying. Under the tiny setting, for both linear and quadratic CausalRNs, we chose $d_h = d_e = 192$. The number of (Linear) CausalRN blocks is 12, totaling 1.34 million parameters. CausalRNs are homogeneous architectures, meaning they exclusively use CausalRN layers without mixing MLP or self-attention layers. Residual connections and pre-normalization are used, following the Transformer architecture pattern. For both linear and quadratic Transformers, we chose $d_h = 4d_e = 768$. We used 12 (Linear) Transformer blocks, totaling 5.33 million parameters. For both Transformers, a sweep for the attention heads were performed in $\{1, 2, 4, 8\}$. We found using a single head gave optimal performance under our choice of $d_e = 192$.

**Initialization**   Following common practices, we initialized all biases to zero and all weights randomly following $\mathcal{N}(0, 0.02)$, except for output layers, which were further divided by the square root of the number of residual connections. Token embeddings and positional embeddings were trainable and randomly initialized following $\mathcal{N}(0, 1)$.

**Optimization**   For optimization, we used Adam with $\beta_1 = 0.9$ and $\beta_2 = 0.999$. We linearly warmed up the learning rate within the first 50 iterations. Without data leakage, we performed a learning rate sweep in $\{1 \times 10^{-5}, 5 \times 10^{-5}, 1 \times 10^{-4}, 5 \times 10^{-4}, 1 \times 10^{-3}\}$, and found $5 \times 10^{-4}$ to be optimal for Transformers and $1 \times 10^{-3}$ to be optimal for CausalRNs. To ensure a meaningful comparison of convergence speed, we chose $5 \times 10^{-4}$ since this was the minimum of both. We note that this choice was optimal for Transformers and suboptimal for CausalRNs.

**Training**   We trained all models on a single NVIDIA A100 GPU using a batch size of 320 for a maximum of 2000 iterations. Each training run can be completed within one and a half hours.

**Evaluation**   For each iteration step, we calculated the cross-entropy loss and average accuracy from 320 online samples to evaluate the models. The accuracy was computed in parallel, not through autoregressive decoding. We note that a 100% parallel accuracy necessarily implies a 100% autoregressive accuracy. Both CausalRNs and Transformers eventually achieves 100% parallel accuracy.

