# OpenReview forum: "Linear-Time Sequence Modeling with MLPs"
_ICLR.cc/2025/Conference — Submitted to ICLR 2025_

### Official Review · Reviewer_ExTo · 2024-10-28

**Soundness:** 3
**Presentation:** 3
**Contribution:** 2
**Rating:** 6
**Confidence:** 3

**Summary:**

The authors present an extension of Relation Networks, dubbed CausalRNs. Authors position CausalRNs as a tool to better understand sequence modeling rather than as a replacement for existing architectures (SSMs, Transformers). They employ clever tricks such as normalization approximation and exponential activation to build CausalRNs. Experiments are conducted on WikiText, CIFAR and a copying task to demonstrate that CausalRNs CAN model sequences albeit not as effectively as Transformers.

**Strengths:**

1. The proposed architecture is novel, interesting and it pushes Relation Networks one step further.
2. They scale better than transformers on copying task according to Figure 6

**Weaknesses:**

1. Imho, a major weakness of paper is a mismatch between their positioning and their experiments - authors wish use CausalRNs to study sequence modeling architectures, but present limited insights/ablations against Transformers. Following are a few examples, hopefully actionable:
1A. Authors claim that CausalRNs are worse than transformers because they do not have multi-head attention, tensor cores, no I/O aware. At least the first two limitations can be applied to Transformers - remove the MHA and train on CPUs, compare against such a transformer.
1B. Related to Figure 6, Authors claim that CausalRNs converge faster than transformer but provide no explanation. This could be interesting because Figure 5 shows a similar phenomenon where Transformer loss is higher. Some interesting questions we could ask: Is the gain in convergence mostly in warm-up, or does it sustain through training (i.e., are Causal RN's somehow more data efficient?)

2. One of the benefits of CausalRNs is being able to stack them/make them deeper, or use residual connections - I'd like to see some ablations to see how these models improve with increasing depth.

**Questions:**

Typos?

1. Eq 8: q_j instead of q_i?

---

> ### Author Response · Authors · 2024-11-26
> **Response to Reviewer ExTo**
>
> We thank the reviewer for their thoughtful feedback and careful reading of our paper. We particularly appreciate the identification of the typo in Eq. 8, where indeed $q_i$ should be $q_j$.
>
> &nbsp;
>
> **Weakness #1**
>
> Thank you for pointing out the potential mismatch between our positioning and our experiments. We are drafting a version that makes our motivation and takeaways clearer.
>
> &nbsp;
>
> **Weakness #1a**
>
> We note that the copying task experiments already uses the single-head transformer.
>
> &nbsp;
>
> **Weakness #1b**
>
> In Figures 5 and 6, while the Quadratic CausalRN surpasses Transformer on the copying task, the observed differences are not significant. We think Transformer and the Quadratic CausalRN are generally on par. We don't think there is a fundamental mechanism that would make one significantly faster than the other. To be clear, we do think there is a fundamental reason for them to be on par, which is elaborated in the paper (infinitely-growing matrix-valued memory).
>
> &nbsp;
>
> **Weakness #2**
>
> In our current experiments, we stacked 12 CausalRN blocks as this is a standard configuration. We believe the quantitative results on language modeling and image classification already show the benefit of stacking blocks and residual connections.
>
> &nbsp;
>
> **Proposed Changes**
>
> We are working on a refined version of the paper incorporating the reviewer's feedback, especially better aligning our positioning and experiments.

---

> > ### Author Response · Authors · 2024-12-03
> >
> > Dear Reviewer,
> >
> > We have submitted a revised version that highlights our contributions and improves the overall quality in the experiment section. If you think we have addressed some of your questions and concerns, please consider raising the rating to reflect that.
> >
> > Thank you!

---

### Official Review · Reviewer_1wCs · 2024-10-29

**Soundness:** 4
**Presentation:** 3
**Contribution:** 1
**Rating:** 5
**Confidence:** 4

**Summary:**

The paper presents Causal Relation Networks (CausalRNs), a novel approach to sequence modeling with MLPs. The key innovation is using exponential activation functions to reduce complexity from quadratic to linear by exploiting distributive properties when mixing sequence elements. The architecture combines dot products between activations with causal masking and normalization. Empirical results are given which show that CausalRNs do not outperform transformers, linear transformers, or Mamba in accuracy or perplexity, but do perform better than linear transformers in terms of handling long context.

**Strengths:**

1. Clear and logical presentation, particularly in deriving CausalRNs from relation networks.
2. Well-motivated choice of exponential mapping and normalization methods.
3. Thorough ablation studies demonstrating necessity of each architectural component for CausalRNs.
4. Clear evaluation methodology in the experimental section.

**Weaknesses:**

1. The novelty of the architecture is not obvious, particularly given its similarity to linear attention mechanisms. The paper doesn't sufficiently differentiate the method from prior work on linear attention variants. The basic equation seems quite similar to equation (9) in the linear transformers paper, where the output at position \(j\) is given by a partial sum up to position \(j\). It would be very useful if the authors could clarify how their work differs from linearized attention.
2. The empirical results show significant limitations. The model consistently achieves higher perplexity than comparable architectures. Meanwhile, no clear computational or memory advantages are demonstrated.
3. Overall, there is no clear demonstration of a useful contribution. Some improved in-context learning capabilities are suggested by Figure 6 and warrant deeper investigation which is not given in the paper. More generally, there are no compelling use cases presented where CausalRNs would be the preferred architecture. The theoretical connections to infinite-width networks are mentioned but not really developed.

**Questions:**

1. What are the concrete benefits and practical use cases where CausalRNs would be preferred over existing architectures?
2. Beyond being an "interesting research object," can the authors suggest ways in which this could advance our theoretical understanding of sequence modeling?
3. How does this architecture fundamentally differ from linear attention mechanisms, and is it a special case of linear attention?

EDIT: After reviewing the authors' response, I'm happy to see they have addressed my question about linear attention. I now agree that it is materially distinct from linear attention. I have raised my rating to reflect that. However, I would still not champion acceptance, since the 'research questions' that the authors raise come off as a somewhat random grab-bag of results--there isn't necessarily a clear scientific question or area of study, just some disconnected results.

---

> ### Author Response · Authors · 2024-11-22
> **Response to Reviewer 1wCs (1/2)**
>
> Thank you for your insightful comments about our paper. We are glad you agree that our approach is logically presented and that our ablation study is thorough. Below, we address each of your concerns and questions.
>
> &nbsp;
>
> ## **Regarding Weakness #1 (Similarity to Linear Attention)**
> CausalRN is not a special case of linear attention [1] or its variants [2]. We will explain from a higher level and then present a mathematical proof.
>
> ***High-level explanation:***
>
> Linear attention [1] maintains a matrix-valued memory state of shape (d, d), where d is the embedding size. CausalRN, however, maintains a vector-valued memory state of an arbitrary length. A typical choice is (4d,), but it can be larger or smaller (see Fig. 4(d)).
>
>
>
>
>
> **Proof that CausalRN cannot be a special case of linear attention [1]:**
>
> Let us start by expressing linear attention and CausalRN in their recurrent forms.
>
> ---
>
> Linear Attention (see Algorithm 1 (forward pass) in [1]):
>
> $$
> \begin{align*}
>     S_i &= S_{i-1} + \phi(k_i)^\top v_i, \newline
>     S_i &= \sum_{j=1}^i \left( \phi(k_j) \otimes v_j \right), \newline
>     o_i &= \phi(q_i) S_i, \newline
>     o_i &= \phi(q_i) \sum_{j=1}^i \left( \phi(k_j) \otimes v_j \right).
> \end{align*}
> $$
>
> ---
>
> CausalRN (see Eq. 9 in our paper):
>
> For the purpose of this discussion, we use the notation in [1], replace $\exp$ with $\phi$, and remove the bias terms $b_{\text{in}}$ in CausalRN.
>
> $$
> \begin{align*}
>     o_i &= \phi(q_i) \odot \sum_{j=1}^i \phi(k_j), \newline
>         &= \phi(q_i) \text{diag}\left( \sum_{j=1}^i \phi(k_j) \right), \newline
>         &= \phi(q_i) \sum_{j=1}^i \text{diag}\left( \phi(k_j) \right).
> \end{align*}
> $$
>
> ---
>
> Notice that when $\phi$ is $\exp$:
>
> - $ \text{rank}\left( \text{diag}\left( \phi(k_j) \right) \right) \equiv d $,
> - $ \text{rank}\left( \phi(k_j) \otimes v_j \right) \equiv 1 $.
>
> Under the mild condition that the embedding size is $ d > 1 $, $ \text{diag}\left( \phi(k_j) \right) $ cannot be rewritten into $ \phi(k_j) \otimes v_j $. Therefore, CausalRN cannot be a special case of linear attention.
>
> **QED**
>
>
>
>
>
>
>
>
>
>
>
>
>
>
>
> ***Other Considerations:***
>
> - For a similar reason, CausalRN is not a special case of gated linear attention [2], even though many linear attention variants are.
> - Quadratic CausalRN cannot be represented as either linear attention or gated linear attention, as the memory state is concatenative and grows with the number of tokens like self-attention.
> - CausalRN uses the exponential function and a bias term, both are not found in linear attention.
>
> ***Conclusion:***
>
> Indeed, the only similarity between CausalRN and linear attention is the use of partial sum to enforce causality. We believe what really sets architectures apart is their motivation and properties, and as we will discuss later, CausalRN is novel and interesting on both fronts. We hope this explanation resolves the reviewer's concern.
>
> &nbsp;
>
> **References:**
>
> [1] Katharopoulos, A., Vyas, A., Pappas, N., & Fleuret, F. (2020). Transformers are RNNs: Fast autoregressive transformers with linear attention. In Proceedings of ICML 2020
>
> [2] Yang, S., Wang, B., Shen, Y., Panda, R., & Kim, Y. (2024). Gated Linear Attention Transformers with Hardware-Efficient Training. In Proceedings of ICML 2024

---

> ### Author Response · Authors · 2024-11-22
> **Response to Reviewer 1wCs (2/2)**
>
> ## **Regarding Weakness #2 and #3 (Low Performance & No Clear Contribution)**
> Our work is a scientific investigation in ML that aims to reveal surprising phenomena and explore new research directions. This type of paper, as elaborated by Nakkiran & Belkin in [3], sits between forming a complete theoretical framework and achieving state-of-the-art empirical results. We hope that the reviewer also recognizes the value of such contributions.
>
> From a theoretical perspective, we contributed several surprising discoveries:
> - The ability to train a quadratic number of shared-weight MLPs in linear time.
> - The second ever architecture to match Transformers on the copying task [4].
> - A naturally emerging gating mechanism. For contemporary architectures, the gating mechanism is usually a deliberate design choice.
>
> We presented a breadth of evaluations covering language modeling, image classification, the copying task, interpretability, and necessary ablation studies. This allowed us to report several interesting phenomena that have not been observed by prior work:
> - The phase change phenomenon of Quadratic CausalRN (Figure 5).
> - Convergence behavior in the State Expansion experiments (Figure 4).
> - Attention-like interpretability results (Figure 7).
>
> Our work opens several promising research directions for future investigation, such as:
> - How does a matrix-valued memory state help with in-context retrieval?
> - What are the roles of exponential functions in sequence modeling?
> - How does the infinite width limit affect the behavior of (Causal) Relation Networks?
>
> These additional new directions are out of scope for this paper, but we agree with the reviewer that these will be worthy research investigations for future work.
>
> Given the extent of the contributions mentioned above, we respectfully ask the reviewer to reconsider the contribution rating.
>
> &nbsp;
>
> ## **Questions**
>
> **1. What are the concrete benefits and practical use cases where CausalRNs would be preferred over existing architectures?**
> - CausalRN presents a new, alternative approach to sequence modeling.
> - CausalRN provides a new perspective to advance our theoretical understanding of sequence modeling.
> - CausalRN is not intended to replace Transformer or its variants in practical settings.
>
> **2. Beyond being an "interesting research object," can the authors suggest ways in which this could advance our theoretical understanding of sequence modeling?**
>
> By comparing and contrasting with existing architectures, we were able to filter out the mechanisms that don’t matter and highlight the mechanisms that do matter:
>
> - Exponential functions are indispensable for the phase change phenomenon to occur on a non-Transformer architecture.
> - We verified on a non-Transformer architecture that a matrix-valued memory state is needed for good in-context retrieval ability.
> - We showed that by increasing the state size, the retrieval ability can be significantly increased. A linear model can approach Transformer-level retrieval with a large state size.
>
> These findings can inform future research on efficient sequence modeling architecture design.
>
> **3. How does this architecture fundamentally differ from linear attention mechanisms, and is it a special case of linear attention?**
>
> Please see our response to Weakness #1. CausalRN is not a special case of linear attention.
>
> &nbsp;
>
> **References:**
>
> [1] Katharopoulos, A., Vyas, A., Pappas, N., & Fleuret, F. (2020). Transformers are RNNs: Fast autoregressive transformers with linear attention. In Proceedings of ICML 2020
>
> [2] Yang, S., Wang, B., Shen, Y., Panda, R., & Kim, Y. (2024). Gated Linear Attention Transformers with Hardware-Efficient Training. In Proceedings of ICML 2024
>
> [3] Nakkiran, P., & Belkin, M. (2022). Incentivizing Empirical Science in Machine Learning: Problems and Proposals. In ML Evaluation Standards Workshop at ICLR 2022.
>
> [4] Jelassi, S., Brandfonbrener, D., Kakade, S. M., & Malach, E. (2024). Repeat After Me: Transformers Are Better Than State Space Models at Copying. In Proceedings of ICML 2024.

---

> > ### Author Response · Authors · 2024-12-03
> >
> > Dear Reviewer,
> >
> > We have submitted a revised version that highlights our contributions and improves the overall quality in the experiment section. If you think we have addressed some of your questions and concerns, please consider raising the rating to reflect that.
> >
> > Thank you!

---

### Official Review · Reviewer_VPTR · 2024-11-01

**Soundness:** 3
**Presentation:** 3
**Contribution:** 3
**Rating:** 8
**Confidence:** 3

**Summary:**

The main contribution of the paper is to extend Relation Networks to be both causal and stackable by providing a novel (to my knowledge) new building block they call Bidirectional Relation Networks.  They compute pairwise functions of previous blocks in a causal manner, with a quadratic time like a normal transformer.  They then go on to show that BiRNs can be computed in linear time by choosing the activation function to be an \exp.

However, this linear time version makes certain tasks (like copying string) harder, so they introduce a quadratic variant where they add pre-activation normalization to the pairwise computations, which makes the computational cost be quadratic again but increases the performance on recall tasks like copying.

They perform 3 ablations to show that the pre-normalization activation is important (that increases the cost to be quadratic), that the post-normalization activation is important and that the exponential function is the best normalization function.  They finally demonstrate reasonable performance on text prediction and image classification.

Other interesting observations include the similarity of the training curves to that of a transformer, and the interpretability of their network in the image domain due to their normalization.

**Strengths:**

Introducing simplified new architectures can help elucidate the performance of the most advanced systems; for example the match between the training behavior of the transformer and their model is very interesting.  Newer, simpler architectures can also be more amenable to theoretical analysis and potentially be more interpretable.  Another interesting future question is if the model can perform ICL like a transformer.

Additionally, new architectural units can also be used in conjunction with existing systems (e.g., alternate with attention units), which may further improve global performance.  Overall, it’s important and useful to explore the space of powerful, simplified models, like the authors have done.

**Weaknesses:**

Performance was quite a bit weaker, and the scaling characteristics are unclear.  Naturally, if the performance had been SOTA that would have been very impressive but that is highly unlikely for a new architecture without more optimization.

There isn't quite enough detail of the entire architecture to be able to easily recreate it in their experiments.  Was this a replacement for the attention layer, was no other layer used then 12 of the BiRN layers?  They mention they can use residual connections - were they used?  What tokenizer was used for the text experiment?  Apologies if this was mentioned somewhere and I just missed it.

Minor corrections
* Equation 8 should be p_i + q_j
* Lines in Figure 3 aren’t red-green colorblind friendly, please change.
* (minor) Be clearer in the experiments when cost is exponential vs. linear.

**Questions:**

* See the weakness section for questions about the setup.
* 3c seems surprising - I got from the text that the advantage of the exp is computation, vs. training time or accuracy.  Any comment on why the exp is more efficient for training?  Would something simple like a bias term help
* Was positional encoding used in the text/image tasks?  In general, there isn't quite enough detail in the
* What was the relative computational cost in the experiments vs Transformer and Mamba?
* I’m confused by the state expansion experiment - clarifying within the paper or an appendix would be useful.

---

> ### Author Response · Authors · 2024-11-28
> **Response to Reviewer VPTR**
>
> Thank you for your insightful comments and careful review of our paper. We are glad you agree that our approach is novel and our discoveries are interesting. We share the view that it is important and useful to explore the space of powerful, simplified models. Below, we address each of your questions.
>
> &nbsp;
>
> **Q: "There isn't quite enough detail of the entire architecture to be able to easily recreate it in their experiments. Was this a replacement for the attention layer, was no other layer used then 12 of the BiRN layers? They mention they can use residual connections - were they used? What tokenizer was used for the text experiment?"**
>
> **A:** We agree. In our newly submitted version, we have addressed these points in the implementation details and Section 4.3 Quantitative Results. To clarify: yes, BiRN/CausalRN is a direct replacement for the attention layer, and no other layers are used. We do apply residual connections. For text experiments, we used an ASCII-based character-level tokenizer with a vocabulary of 128.
>
> &nbsp;
>
> **Q: "Minor corrections."**
>
> **A:** Thank you for your careful reading! In our recently submitted version, we have corrected the typo in Equation 8, updated Figure 3 to be red-green colorblind friendly, and clarified which experiments have quadratic versus linear time complexity. If Figure 3 is still not accessible enough, please give us direct suggestions and we will implement them in our camera-ready version.
>
> &nbsp;
>
> **Q: "3c seems surprising - I got from the text that the advantage of the exp is computation, vs. training time or accuracy. Any comment on why the exp is more efficient for training? Would something simple like a bias term help"**
>
> **A:** To our knowledge, the theoretical explanation for why softmax or exponential functions appear indispensable in modern sequence modeling architectures remains an open question, and we welcome future research on this topic. One possible explanation is that exp(x) enables important tokens to dominate the feature vector after the reduction step. We have clarified this point in the new version.
>
> &nbsp;
>
> **Q: "Was positional encoding used in the text/image tasks?"**
>
> **A:** All experiments used a randomly initialized and trainable positional embedding scheme. While this was mentioned in the implementation details section, we have made it more explicit in our new version.
>
> &nbsp;
>
> **Q: "What was the relative computational cost in the experiments vs Transformer and Mamba?"**
>
> **A:** In our experiments, Linear CausalRNs are computationally lighter than Transformers and comparable to Mamba. The quadratic CausalRN, since it is not I/O-aware and requires O(n²) storage in VRAM, is significantly more resource-intensive than Transformers with Flash Attention. Nevertheless, the quadratic CausalRN remains computationally feasible. We were able to finish training a quadratic CausalRN with a context window of 512 within one and a half hour on the copying task.
>
> &nbsp;
>
> **Q: "I’m confused by the state expansion experiment - clarifying within the paper or an appendix would be useful."**
>
> **A:** We elaborated the state expansion experiment with more detail in the paper. In essence, we increased the number of hidden neurons in the MLPs and measured how it affects the memory capacity of CausalRN. We observed that, as the number of hidden neurons increases, the model's memorization capacity also grows. We observe an early convergence phenomenon, suggesting sequence modeling architectures may not require infinite memory states to achieve perfect retrieval.

---

> > ### Comment · Reviewer_VPTR · 2024-12-02
> >
> > Thank you for your useful clarifications and improvements to the paper.  I'll leave the original score unchanged.

---

### Comment · Area_Chair_SVRB · 2024-12-02
**Discussions between reviewers and authors**

Time for discussions as author feedback is in. I encourage all the reviewers to reply. You should treat the paper that you're reviewing in the same way as you'd like your submission to be treated :)

---

### Meta-Review · Area_Chair_SVRB · 2024-12-21

**Metareview:**

This paper proposes an extension to relational networks. The idea is to compute pairwise functions of previous blocks in a "causal", then the authors provided two versions of the network, one can be computed in linear time (like RNNs) and another still keeping the quadratic time computation but with more normalisation layers.

While the reviewers think the idea is novel and interesting (especially the linear-time option is different from Linear Attention), the major issue of the paper is that the Linear-time version of the proposed network under-performs, where the gap between the propose method and existing Transformers (and Linear Transformers) is not small. This fact is also acknowledged by Reviewer VPTR who gave the highest score of the paper.

After a brief read, I think the approach is interesting and novel, however the paper needs to re-position itself, or the authors need to reconsider their research direction. CausalRN seems promising but the linear-time version of it (Linear CausalRN) is, unfortunately, lacking behind. This contradicts with the title and the motivation of the paper (I'm guessing the authors would like to win over Transformers and linear SSM style models), however, the CausalRN idea itself can still be interesting within the direction of relational networks.

I encourage the authors to revise their manuscript based on the feedback from this reviewing process.

**Additional Comments On Reviewer Discussion:**

AC - reviewer discussion received further comments, see below.

"
As I wrote in my update, the authors addressed my question about whether their mechanism is merely a variant of linear attention. I would still be on the side of leaning rejection, since it's not really obvious to me what the true contribution is for the paper. The authors say that the point is to answer 'research questions', but the 'research questions' that the authors raise come off as a somewhat random grab-bag of results--there isn't necessarily a clear scientific question or area of study, just some disconnected results.
"

---

### Decision · Program_Chairs · 2025-01-22

Reject